# Volumetric Flow Assessment in Extracranial Arteries in Patients with 70–99% Internal Carotid Artery Stenosis

**DOI:** 10.3390/diagnostics12092216

**Published:** 2022-09-13

**Authors:** Piotr Kaszczewski, Michał Elwertowski, Jerzy Leszczyński, Tomasz Ostrowski, Joanna Kaszczewska, Tomasz Brzeziński, Daniel Jarosz, Siavash Świeczkowski-Feiz, Zbigniew Gałązka

**Affiliations:** Department of General, Endocrine and Vascular Surgery, Medical University of Warsaw, Banacha 1A, 02-091 Warsaw, Poland

**Keywords:** Doppler ultrasonography, carotid stenosis, carotid artery disease, TIA, stroke, cerebral blood flow, cerebrovascular reserve

## Abstract

Background: Cerebral blood flow (CBF) can be related to the risk of occurrence of neurological symptoms. Well-developed collateral circulation is a good prognostic factor in patients with cerebrovascular disease. Understanding the mechanisms of collateral circulation may be important in the diagnosis, treatment, and monitoring in this group of patients. The aim of this study covered the assessment of CBF in patients with 70–99% Internal carotid artery (ICA) stenosis, focusing on the circulation pathways and flow volume in extracranial arteries. Materials and methods: 53 patients with 70–99% ICA stenosis (mean age 73.4 ± 7 years old; 17 female, mean age 73.9 ± 7.5 years old; 36 male, mean age 73.2 ± 6.8 years old) were included in the study. In all patients a Doppler ultrasound (DUS) examination, measuring blood flow volume in the internal carotid artery (ICA), external carotid artery (ECA), and vertebral artery (VA), was performed. The cerebral blood flow (CBF) was compared to the previously reported CBF values in the healthy population > 65 years old. Results: Among the study groups three subgroups with flow differences were identified: patients with elevated CBF (significant volumetric flow compensation—26/53, 49%), patients with CBF similar to (mild compensation—17/53, 32%), and patients with CBF lower than (no compensation—10/53, 19%) the healthy, equally aged population. The percentage of patients with significant volumetric flow compensation was the highest in age groups 65–69 years old (62.5%) and >80 years old (60%). In the oldest age group (>80 years old) no patients without flow compensation (0%) were observed. The level of compensation depends on the number of the arteries with compensatory increased flow. In patients with significant volumetric flow compensation, the compensatory increased flow was observed, on average, in 3.31 arteries, in the mild compensation group—in 2.18 arteries, and in the no compensation group only in 1 artery. ICA plays the most important role in the volumetric flow compensation—the increase in the flow volume, in comparison to the reference values, was between 116.7 mL/min and 251.9 mL/min (in the ECA 57.6 mL/min–110.4 mL/min; in the VA 73.9 mL/min–104.9 mL/min). The relative flow increase was highest in the VA: 215–246%, then in the ECA: 163–206%, and finally in the ICA: 148.6–192%. The increased flow was most commonly observed in the VA—57 arteries, in second place in the ECA—42 arteries, and ICA—31 arteries. In patients with unilateral ICA stenosis, the volumetric flow increase was stated more frequently in the ipsilateral ECAs then in the contralateral ones (23 vs. 14). In the VA the opposite tendency was observed (29 contralateral vs. 23 ipsilateral). The ability of volumetric flow compensation decreased significantly with increasing age. Conclusions: Understanding the mechanisms of collateral circulation and their assessment in Doppler ultrasonography may provide a novel and easily accessible tool of identifying and monitoring patients with cerebrovascular disease.

## 1. Introduction

Atherosclerosis affects about 20% of population above 60 years old and more than 50% aged over 85 years old, and it is becoming a global health burden due to the aging of the population [1,2].

Stroke, together with myocardial infraction are the leading causes of deaths due to cardiovascular diseases. In 2019 the global prevalence of stroke was 101.5 million people, out of whom over 6.6 million died [3]. Ischaemic stroke accounts for about 85% all strokes, while up to 20% of these events may be secondary to carotid artery atherosclerotic disease [4].

Cerebral blood flow (CBF) correlating with cerebrovascular reserve (CVR) is associated with and may be useful in predicting both: the risk of occurrence of ischaemic symptoms and their clinical outcome. It has been proved that well developed collateral circulation is linked to smaller stroke size and better reperfusion treatment and rehabilitation outcomes [5,6,7,8].

Recently a series of works concerning volumetric flow assessment in extracranial arteries in Doppler Ultrasonography in patients with carotid artery disease were published. It was shown that among patients with carotid artery stenosis and occlusion, there are subgroups with CBF changes. In patients with severe ICA stenosis or occlusion there are individuals in whom the CBF, despite the presence of significant stenosis of one of the extracranial vessels, is elevated compared to healthy equally aged population, due to a significant increase in the flow in other extracranial arteries. There are also patients in whom the CBF is similar to healthy population—due to the mild increase in the flow in other extracranial vessels, and the patients in whom there is no flow increase in other arteries—those patients have lower CBF values than the healthy population. It was reported that higher CBF values are correlated with a lower risk of occurrence of ischaemic symptoms, while in patients with lower CBF values the risk of occurrence of ischaemic symptoms is twice as large [9,10,11,12,13,14].

The aim of this study was to assess the compensatory mechanisms with Doppler ultrasonography, examining the circulation pathways and flow volume in patent extracranial arteries, in a group of patients over 65 with internal carotid artery stenosis of 70–99%.

## 2. Materials and Methods

The study group included 53 patients with ICA stenosis 70–99% (mean age of group 73.4 ± 7 years old; 17 female, mean age 73.9 ± 7.5 years old; 36 male, mean age 73.2 ± 6.8 years old). Detailed data are presented in Table 1.

The flow volume values, which in this study are hereafter called “reference” values or the “proposed reference standard,” were established by our team based on a group of 123 healthy volunteers (without concomitant disorders, which could influence CBF values) and published in 2020. The reference range is presented as the average flow volume (mL/min) ± standard deviation (mL/min); see Table 2 [10].

In this study, the flow volumes in extracranial arteries (CCA, ECA, ICA, and VA) in patients with ICA occlusion were compared with those obtained for a group of healthy volunteers in order to determine the pathways and the degree of compensatory circulation.

In this study, values exceeding the proposed reference values (see Table 2) for average + standard deviation are referred as to having “compensatory increased flow” or “significant compensation.”

The values within the proposed reference are referred to as “Flow volume within reference value” or “mild compensation”—in the presence of a major reduction in flow in one of the carotid arteries, an increase in the other vessels allows it to maintain the CBF within the proposed standards.

A blood flow volume lower than the proposed reference value for average and standard deviation is referred as to “no compensation” or “decreased flow volume.”

Recruitment of patients to both groups in terms of concomitant disorders was conducted according to the previously described protocol, in order to eliminate its influence on cerebral blood flow volume, except for ICA stenosis/occlusion [10].

In all patients, a DUS examination measuring blood flow volume in the common carotid arteries (CCAs), patent internal carotid artery (ICA), external carotid arteries (ECAs), and vertebral arteries (VAs) was performed. The flow volume in the CCA was measured as a control; the measurements were considered accurate when the sum of the flow volumes in the ipsilateral ICA and ECA (measured distally to the superior thyroid artery), or ECA only (in the case of ipsilateral ICA obstruction), was slightly lower than the flow volume in the CCA (due to flow volume loss in the superior thyroid artery).

The cerebral blood flow (CBF) was calculated as the grand total of the flow volumes in all the aforementioned patent extracranial arteries: ICA, ECAs—distal to the origin of the superior thyroid artery, and VAs.

Examinations were conducted following a previously described protocol [5], by the same experienced sonographer, using a Canon Aplio i800 ultrasound scanner with Linear i11LX3 transducer (Canon Medical Systems Corporation, Otawara, Tochigi, Japan). The blood flow volumes were calculated using the ultrasound scanner’s semiautomatic program.

The diameter of each vessel was measured using three different techniques: B-mode, SMI (superb microvascular imaging) mode, and B-mode combined with SMI image. All measurements (both diameter measurement and volumetric assessment) were carried out three times and their average was considered the final result.

The stenosis degree assessment was based on the Results of Society of Radiologists in Ultrasound Consensus Conference on the diagnosis of Internal Carotid Artery Stenosis and DEGUM criteria—using the peak systolic velocity (PSV), end-diastolic velocity (EDV) changes as well as ICA PSV/CCA PSV ratio, post-stenotic flow disturbances (severity and length), and reduction. Additional criteria were used for ICA stenoses that exceeded 80%: prolongation of acceleration time >0.4 s and flow velocity reduction in the upper portion of the vessel and the PSV ratio between the stenosis and the upper part over 8.0 [9,12,15,16,17].

### Statistical Analysis

Statistical analysis was performed with Statistica 13 (StatSoft Polska Sp. z o.o., Krakow, Poland).

For the comparison of the two groups, the t-test and the Mann–Whitney U test were used. The Shapiro–Wilk test was performed as a test of normality. Levene’s test was used to assess the equality of variances. The normal distribution of data with equal variances was a prerequisite to use the t-test. With no equality of variances, the t-test with Cochran–Cox correction was performed. When one of the variables had an abnormal distribution, a nonparametric Mann–Whitney U test was performed. The results were considered statistically significant when the *p*-value was below 0.05. Additionally, a Bonferroni correction for multiple comparisons was performed in order to reduce the risk of type I errors; the significance level was set to 0.005.

Additionally, a linear regression analysis was performed. The correlation was considered statistically significant when the *p*-value was below 0.05.

## 3. Results

### 3.1. Cerebral Blood Flow Volume in the Whole Study Group (53 Patients)

Among the study group (53 patients) the three subgroups with CBF differences were identified:26/53, 49%—patients with significant volumetric flow compensation (CBF values exceeding the proposed reference values: average + standard deviation);17/53, 32%—patients with mild compensation (in the presence of major reduction in the flow in one of carotid arteries, the increase in the other vessels allows the CBF to be maintained within the proposed standards);10/53, 19%—patients with no compensation (CBF lower than the proposed reference value: average—standard deviation).

The percentage of patients with “significant compensation” was the highest in the age group 65–69—62.5%. It decreased to 29.4% in the age group 70–74 years old, and was continuously raised from 50% in the age group 75–79 years old to 60% in the oldest age group ≥ 80 years old.

The percentage of patients with mild compensation showed slight fluctuations from 20% (age group 75–79 years old) to 41.2% (age group 70–74 years old). In the youngest age group, 65–69 years old, there were 25% of patients with “mild compensation” while in the oldest age group, ≥80 years old, they constituted 40% of the group.

Patients with “no compensation” tended to be the least prevalent group (except age group 75–79). This group accounted for 12.5% of patients aged 65–69 years old, 29.4% of the group aged 70–74 years old, and 30% of patients aged 75–79 years old. In the oldest age group (≥80 years old) no patients “without compensation” were observed.

Detailed data concerning the flow compensation in the whole study group are presented in Table 3 and Figure 1.

### 3.2. Cerebral Blood Flow Volume in the Whole Study Group (53 Patients)

ICA plays the most important role in the volumetric flow compensation—the increase in the flow volume, in comparison to the reference values, was between 116.7 mL/min and 251.9 mL/min (in the ECA 57.6 mL/min–110.4 mL/min; in the VA 73.9 mL/min–104.9 mL/min).

The relative flow increase was highest in the VA: 215–246%, then in the ECA: 163–206%: and finally in the ICA: 148.6–192%.

The data concerning the level of compensation in all extracranial arteries with increased flow volume are presented in Table 4 and Figure 2.

### 3.3. The Pathways of Volumetric Flow Compensation in the Extracranial Arteries

The data concerning the pathways of collateral circulation in patients with unilateral ICA occlusion are presented in Table 5, and with bilateral ICA occlusion in Table 6.

Volumetric flow compensation in patients with unilateral ICA occlusion was most frequently observed in:31 contralateral ICA;29 contralateral VA;23 ipsilateral ECA;23 ipsilateral VA;14 contralateral ECA.

Altogether in the whole study group the compensatory increased flow volume was observed in the 31.13% of all ICAs, 39.62% of all ECAs, and 51.89% of all VAs. The detailed percentages of the ICA and ECA, and data concerning the percentage of arteries with flow compensation are presented in Figure 3.

In patients with unilateral carotid stenosis, the compensatory increased flow volume was more frequently observed in ECAs in the ipsilateral side (23 vs. 14), and in the contralateral side in VAs (29 vs. 23)—see Figure 4.

### 3.4. The Multivessel Character of Volumetric Flow Compensation

The character of volumetric flow compensation is multivessel. In the group with “significant volumetric flow compensation”, compensatory elevated blood flow volume was observed in 2–5 arteries—on average in 3.31 vessels. The median value was 3.

In the group with “mild compensation”, compensatory elevated blood flow volume was observed in 1–4 arteries—on average in 2.18 vessels. The median value was 2.

In the group with “no compensation”, compensatory elevated blood flow volume was observed in 0–3 arteries—on average in 1 vessel. The median value was 1.

The data are presented in the Figure 5.

### 3.5. The Correlation between Volumetric Flow Compensation, Cerebral Blood Flow Volume, and Age

Analysing all the arteries from the whole study group, a non-significant, slightly decreasing correlation between blood flow volume in the ICA, ECA, and VA was observed—see Figure 6A–C.

There was a significant decreasing correlation between the CBF and increasing age (*p* = 0.0002)—see Figure 6D.

When analysing only vessels with compensatory increased flow, a statistically significant, decreasing corelation was observed between flow volume in the ICA, ECA, and VA and age—see Figure 7A–C, respectively. This explains the decreasing correlation between CBF and increasing age—see Figure 6D or Figure 7D.

## 4. Discussion

The brain is a highly active and energy-consuming organ. While its mass is estimated to be only 2% of human body weight, its metabolism requires one fifth of the body’s oxygen, comprising almost 18% of cardiac output to generate a perfusion of 50 mL/100 g/min, with higher flow in grey matter of 80 mL/100 g/min and lower in white matter of 20 mL/100 g/min [18,19,20].

In the physiological aging process the brain mass gradually declines. The process begins after exceeding the age of forty, initially at the rate of 5% per year, increasing over 70 years old [19,20,21].

This process is accompanied by corresponding changes in brain perfusion and CBF. The annual cerebral perfusion decline rate is reported for 3 mL/min/year to 4, 8 mL/min/year [22,23,24]. Recently, based on the CBF assessment in Doppler ultrasonography in the group of 179 healthy volunteers aged 18–96 years old, the annual decline in the CBF was reported to be 7.6 mL/year. In the group aged below 65 years old, the annual decline in cerebral blood flow (CBF) was lower and estimated to be 6.2 mL/year. It became more prominent with age, reaching 11.4 mL/year in people aged 65–75 years old, increasing to 14.35 mL/year above 75 years old [10]. The greater values of the CBF decline may result from the different CBF quantification method—the authors quantified CBF as a grand total of the flow in all extracranial arteries ICAs, VAs, and ECAs, which was included in the measurement because of its vital role in collateral circulation in patients with severe ICA stenoses.

In this study, patients with an ICA stenosis of 70–99% were included. Within this group, three subgroups with flow differences were identified patients with “significant volumetric flow compensation” (with CBF higher than the healthy, equally aged population), patients with “mild compensations” (CBF equal to healthy equally aged people) and patients with “no compensation” with CBF lower than comparable healthy peers. Despite the differences in the haemodynamics caused by severe ICA stenosis, a statistically significant flow decline in the CBF of 10.1 mL/min/year was observed in the whole study group. There was also a discrete, non-significant decline in the flow volumes in the ICA, ECA, and VA, which in consequence resulted in a significant CBF decrease.

In the study group, there were however arteries in which the flow volume increased significantly—arteries with volumetric flow compensation. When analyzing only arteries with flow compensation, a statistically significant flow volume decline was observed in every extracranial artery: ICA, ECA, and VA. The fact that the ability of volumetric flow compensation decreased with age is in accordance with contemporary knowledge concerning brain aging process. The mass of brain declines and CBF decreases; therefore, the ability of the brain to increase the flow may be worse with age.

Interesting observations can be made when comparing to the group of patients with ICA occlusion [13]. In patients with ICA occlusion there was no physiological decrease in the CBF—only non-significant decreased tendency. The ability of compensation in ICA seemed to increase non-significantly with age, while a small declining, still not significant tendency was observed in the ECA and VA. What is common in these studies is the fact that in the eldest age group, ≥80 years old, there were no patients without compensation, while the “significant compensation” group was most prevalent [13]. This may suggest that proper blood supply to the brain—CBF, facilitated by compensatory increase in the flow volume in one or more extracranial arteries promotes and facilitates patient survival.

Cerebral blood flow is closely associated with cerebrovascular reserve CVR, and therefore with the risk of forthcoming ischemic events, regardless of the presence of neurological symptoms, stenosis, or occlusion of the supplying artery, or the cerebrovascular reactivity testing method [25].

It has recently become apparent that collateral circulation plays an important role as a reliable and sensitive predictor of the occurrence of ischaemic symptoms (including stroke), their severity, and clinical outcomes of the treatment or rehabilitation [5,6,7,8].

The significant compensatory elevation of CBF is more frequently observed in asymptomatic patients suffering from severe ICA stenosis, which suggests its protective influence on developing ischaemic symptoms. The relative risk of observing the compensatory increased flow in extracranial arteries in asymptomatic patients is almost two times higher than in symptomatic ones [9,12].

In our study we observed indirectly the mechanisms of the recruitment of the collateral circulation. It was achieved by increasing the flow volume in one or more extracranial arteries. The number of the arteries with compensatory increased flow varied between the groups with different compensatory status. In the group with “significant volumetric flow compensation” compensatory elevated blood flow volume was observed in 2–5 arteries—on average in 3.31 vessels and in the “mild compensation” group in 1–4 arteries—on average in 2.18 vessels. In the group with “no compensation” group compensatory elevated blood flow volume was observed in 0–3 arteries—on average in 1 vessel.

It is generally thought that the Circle of Willis (CoW) is most important anastomosis in case of ICA or VA stenosis, connecting vascular beds of different extracranial arteries and facilitating collateral circulation. Its role in the development of the TIA or stroke seems obvious; however, there are contradictory opinions and data in the literature. Several studies have demonstrated a significant correlation between abnormalities in the CoW and ischaemic stroke [26,27,28,29,30,31].

The meta-analysis of Oumer et al. on 2718 patients did not demonstrate a statistically significant correlation between CoW morphology and stroke (just a non-significant positive association) [32].

Jongen et al. in a group of patients with increasing ICA stenosis, observed a gradual decrease in CBF, independent of the CoW morphology. Authors explained this fact with other collateral pathways, including ophthalmic and leptomeningeal vessels, which may compensate for the CoW collaterals [33].

Myrcha et al. suggested that the anatomy of the Circle of Willis itself is not a strong predictive factor for the prognosis of cross-clamping intolerance [34].

Research concerning cerebral vasculature anatomy has reported that CoW is incomplete in a large percentage of the population—physiological configuration is identified in 14.2–72.2% of patients, depending on the study [35,36,37].

Some authors have doubted the compensatory function of the Circle of Willis, stressing the small size of the arteries and the large percentage of anatomic abnormalities in the CoW, suggesting that it may serve as a passive energy (pressure)-dissipating system, transferring pressure without considerable blood flow from the high-pressure end to the low-pressure end, where the pulse wave and blood flow arrive asynchronously [38].

Neurocognitive functioning is another important aspect strongly correlated to CVR and therefore also with CBF. Lattanzi et al. observed the improvement of cognitive performance after carotid endarterectomy (CEA) in patients with TIA and ipsilateral high-grade ICA, who initially had decreased values of CVR. The improvement correlated inversely with age and preoperative CVR values [39,40].

In this study the compensation had a multivessel character. ICA played the most important role in the volumetric flow compensation—the increase in the flow volume, in comparison to the reference values, was between 116.7 mL/min and 251.9 mL/min (in the ECA 57.6 mL/min–110.4 mL/min; in the VA 73.9 mL/min–104.9 mL/min). The relative flow increase was highest in the VA 215–246%, then in the ECA 163–206% and finally in the ICA: 148.6–192%. Similar tendencies were observed with patients with ICA occlusion; however, the relative flow increase was higher there, up to 199.8% in the ICA, 222.5% in the ECA, and 389.6% in the VA [13].

The compensatory increased flow volume was observed in this study: 51.89% of VAs, 39.62% of ECAs, and 31.13% of ICAs. In patients with ICA occlusion the flow compensation was observed in about 50% of all patent ICAs (54.76%), ECAs (48.91%) and VAs (51.09%) [13].

In patients with unilateral ICA stenosis, the volumetric flow increase was stated more frequently in the ipsilateral ECAs then in the contralateral ones (23 vs. 14). In the VA the opposite tendency was observed (29 contralateral vs. 23 ipsilateral). Comparing this data to the patients with ICA occlusion, the compensatory increased flow was observed more commonly in ipsilateral vessels both in the ECA and VA (23 vs. 16—ECA and 22 vs. 17—VA) [13].

Our study has limitations. Using only Doppler ultrasonography to examine extracranial vessels, we were not able to determine the anatomy of intracranial vessels. We assessed the hemodynamic processes in the central nervous system indirectly. The main advantage of our method is its availability—it can be performed even in an outpatient clinic with a medium class ultrasound scanner.

We would also like to refer to the “reference values” which in our studies were initially defined within one standard deviation from the mean. During the preliminary research our team identified a cut-off point in almost exactly one sigma from the mean. Not only did it allow us to identify the differences between asymptomatic and symptomatic patients but it also permitted a prediction of the postoperative increase in CBF in patients who undergo surgical treatment due to ICA stenosis [12,41]. Narrowing the confidence interval is acceptable when there are justified indications.

## 5. Conclusions

Volumetric flow compensation has a multivessel character in which the ICA plays the most important role. The ECA and VA are also key importance pathways, especially in severe ICA stenosis or occlusion.

The compensatory status of the patients depends mainly on the number of the extracranial arteries with increased blood flow.

The ability of volumetric flow compensation decreases gradually with age.

Volumetric flow assessment in Doppler ultrasonography may provide an easily accessible tool for diagnosing and monitoring patients with cerebrovascular disorders.

## Figures and Tables

**Figure 1 diagnostics-12-02216-f001:**
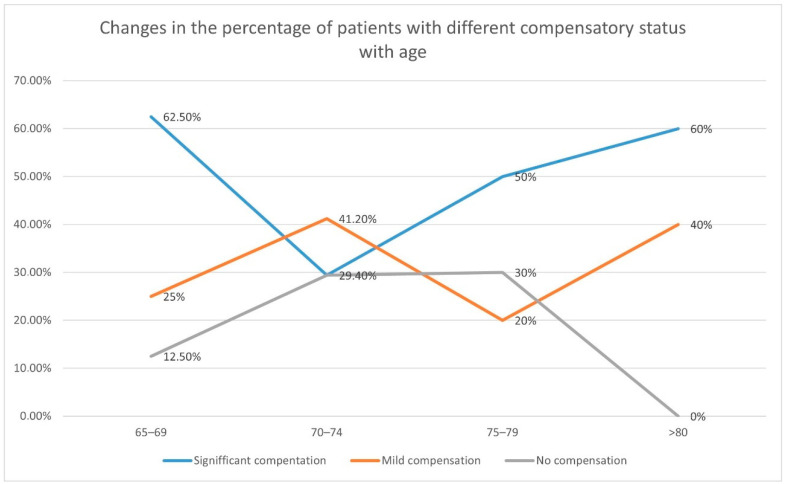
Tendencies in CBF changes in different age group. The percentage of patients with significant flow compensation tended to rise with increasing age. The percentage of patients with mild compensation did not change significantly, showing only slight fluctuations from 20% to 41.2%. The percentage of patients without compensation tended to be least prevalent group. These patients were not observed in the oldest age group.

**Figure 2 diagnostics-12-02216-f002:**
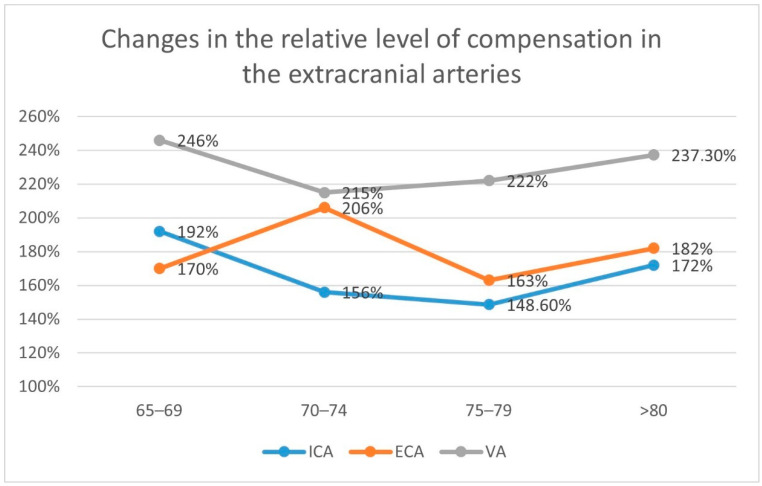
Changes in the relative level of compensation in extracranial arteries in patients with ICA occlusion. No significant increasing or decreasing tendencies or correlations concerning the level of compensation were observed in the ICA, ECA, and VA.

**Figure 3 diagnostics-12-02216-f003:**
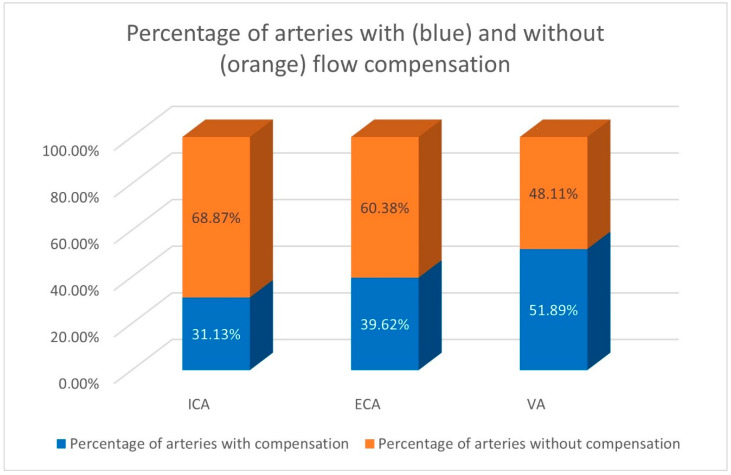
The percentage of the arteries with and without flow compensation in the whole study group. The compensatory increased flow volume was observed in 51.89% of VAs, 39.62% of ECAs, and 31.13% of ICAs.

**Figure 4 diagnostics-12-02216-f004:**
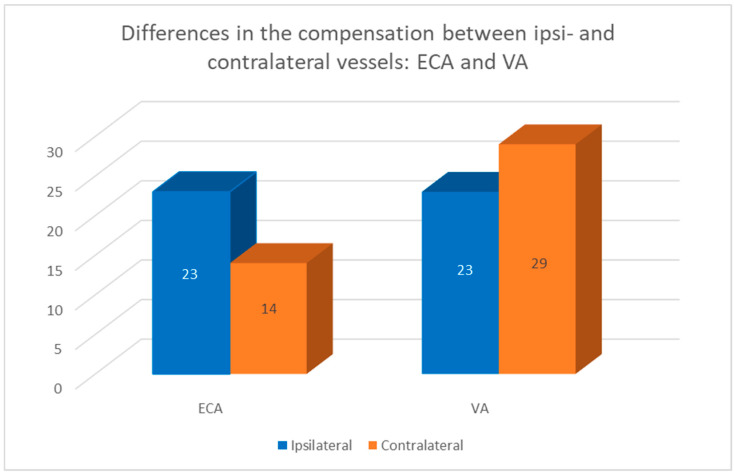
Differences in compensation in the ECA and VA between ipsilateral and contralateral vessels. Compensatory increased flow was more frequently observed in ipsilateral ECA (23 vs. 14) and in contralateral VAs (29 vs. 17).

**Figure 5 diagnostics-12-02216-f005:**
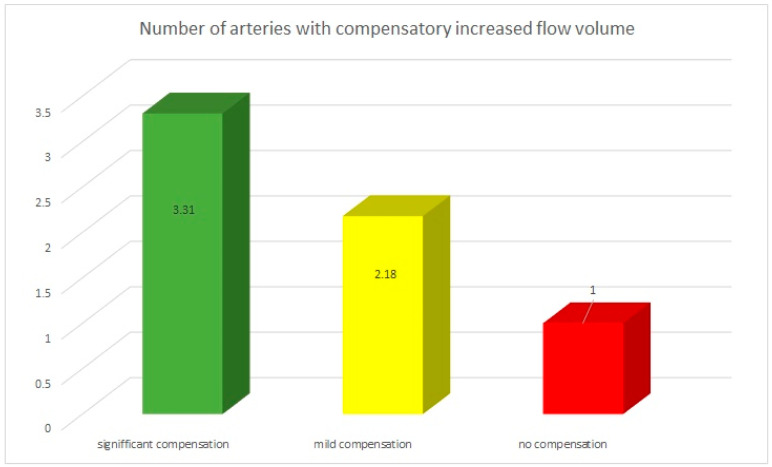
Average number of arteries with compensatory increased flow volume involved in the volumetric flow compensation. In the “significant compensation group” the compensatory elevated blood flow volume was observed in 3.31 vessels, in the “mild compensation” group in 2.18 vessels, and in the “no compensation” group only in 1 vessel.

**Figure 6 diagnostics-12-02216-f006:**
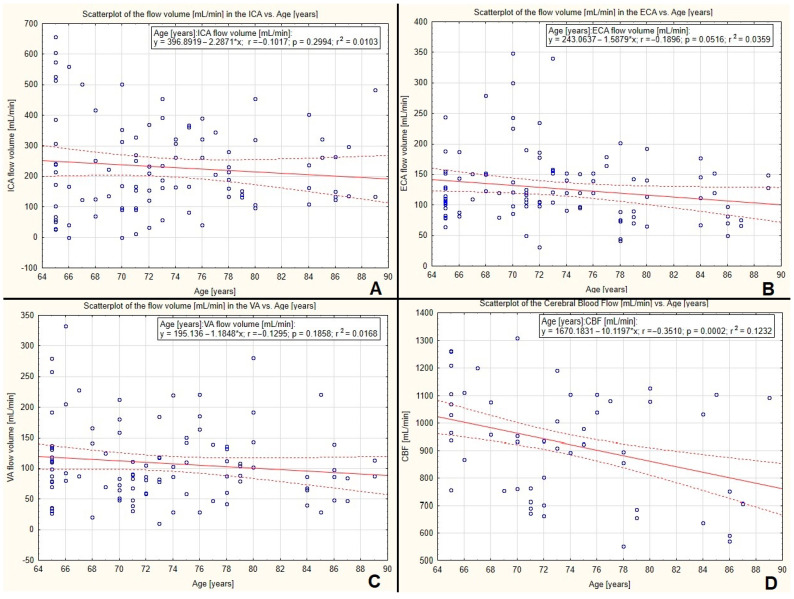
Regression analysis between blood flow volume in extracranial vessels: ICA (**A**), ECA (**B**), VA (**C**), and CBF (**D**) with age. A slightly decreasing, non-significant correlation between blood flow volume in the ICA, ECA, and VA was observed—(**A**–**C**), respectively. Statistically significant decreasing correlation between the CBF and increasing age (*p* = 0.0002) was noted—(**D**).

**Figure 7 diagnostics-12-02216-f007:**
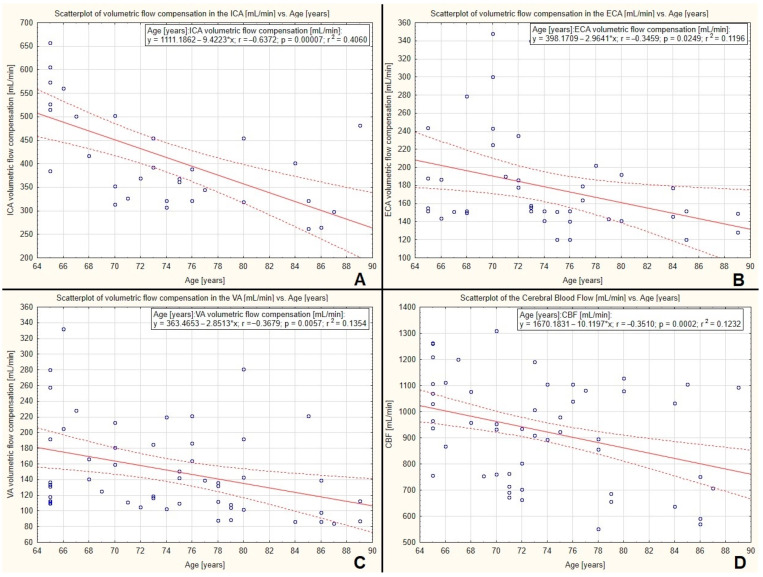
Regression analysis between blood flow volume in extracranial vessels with compensatory increased flow: ICA (**A**), ECA (**B**), VA (**C**), and CBF (**D**) with age. A significant decreasing correlation between blood flow volume in the ICA, ECA, and VA was observed—(**A**–**C**), respectively. This may explain the significant decreasing correlation between the CBF and increasing age (*p* = 0.0002), which is presented here once again—(**D**) (also Figure 6D).

**Table 1 diagnostics-12-02216-t001:** Study group characteristics.

	Female	Male	Total
**Number of patients**	17	36	53
**Mean age** **±** **std. dev.**	73.9 ± 7.5 years old	73.2 ± 6.8 years old	73.4 ± 7 years old
**Number of patients with LICA stenosis**	5	11	16
**Number of patients with RICA stenosis**	10	21	31
**Number of patients with bilateral ICA stenosis**	2	4	6
**Age: 65–69**	5	11	16
**Age: 70–74**	5	12	17
**Age: 75–79**	3	7	10
**Age: 80+**	4	6	10

**Table 2 diagnostics-12-02216-t002:** Reference flow volume in extracranial arteries RICA, LICA—right and left internal carotid artery. RECA, LECA—right and left external carotid artery. RVA, LVA—right and left vertebral artery. Values are presented as the average ± standard deviation.

Age Group	65–69	70–74	75–80	>80
CBF Proposed reference value [mL/min]	898.5 ± 119.1	838.5 ± 148.9	805.1 ± 99.3	685.7 ± 112.3
RICA [mL/min]	271.1 ± 63.6	236.0 ± 66.1	234.8 ± 62.3	202.3 ± 38.4
RECA [mL/min]	106.1 ± 35.0	103.7 ± 33.2	94.0 ± 24.14	83.1 ± 36.3
RVA [mL/min]	58.7 ± 29.1	60.2 ± 26.7	62.3 ± 28.4	55.7 ± 24.1
LICA [mL/min]	276.4 ± 57.5	239.8 ± 42.4	245.5 ± 32.3	204.4 ± 47.0
LECA [mL/min]	101.4 ± 30.9	104.7 ± 32.5	89.0 ± 21.9	79.0 ± 33.7
LVA [mL/min]	84.9 ± 33.0	80.4 ± 29.8	70.0 ± 21.5	58.8 ± 13.0

Adapted from Ref. [10]: Copyright Year 2020, Kaszczewski P; Elwertowski, M.; Leszczynski, J.; Ostrowski, T.; Galazka, Z.

**Table 3 diagnostics-12-02216-t003:** Detailed data concerning the flow compensation in different age groups.

Age/Flow Compensation	Significant Compensation	Mild Compensation—Flow Volume within Reference Value	No Compensation—Decreased Flow Volume
**Whole study group**	26/53 (49%)	17/53 (32%)	10/53 (19%)
**65–69**	10/16 (62.5%)	4/16 (25%)	2/16 (12.5%)
**70–74**	5/17 (29.4%)	7/17 (41.2%)	5/17 (29.4%)
**75–79**	5/10 (50%)	2/10 (20%)	3/10 (30%)
**≥** **80**	6/10 (60%)	4/10 (40%)	0/10 (0%)

**Table 4 diagnostics-12-02216-t004:** The level of compensation in all extracranial arteries.

Age Group	Artery	Compensation Average ± std. dev. [mL/min]	Reference [mL/min]	*p* < 0.005	Flow Difference [mL/min]	Relative Flow Increase
**65–69**	ICA	525.7 ± 81.4	273.8 ± 60.5	yes	251.9	192%
	ECA	176.8 ± 50.3	103.6 ± 32.9	yes	73.8	170%
	VA	176.7 ± 68.1	71.8 ± 32.3	yes	104.9	246%
**70–74**	ICA	370.9 ± 67.7	237.9 ± 54.3	yes	133	156%
	ECA	214.6 ± 70.6	104.2 ± 32.7	yes	110.4	206%
	VA	151.3 ± 45.9	70.3 ± 28.8	yes	81	215%
**75–79**	ICA	356.8 ± 25.5	240.1 ± 47.3	yes	116.7	148.60%
	ECA	149.1 ± 27.1	91.5 ± 23	yes	57.6	163%
	VA	134.4 ± 37.6	60.5 ± 25	yes	73.9	222%
**>80**	ICA	350.4 ± 85.2	203.2 ± 42.7	yes	147.2	172.00%
	ECA	150.6 ± 23.8	81 ± 35	yes	69.6	186%
	VA	136.0 ± 63.7	57.3 ± 18.5	yes	78.7	237.30%

**Table 5 diagnostics-12-02216-t005:** Pathways of collateral circulation in patients with unilateral ICA occlusion.

Age Group	Number of Patients	Stenosis	Contralateral ICA	Contralateral ECA	Contralateral VA	Ipsilateral ECA	Ipsilateral VA
**65–69**	6	RICA	5/6 (83%)	0/6 (0%)	6/6 (100%)	2/6 (33%)	2/6 (33%)
	7	LICA	6/7 (86%)	1/7 (14%)	3/7 (43%)	4/7 (57%)	6/7 (86%)
**70–74**	14	RICA	8/14 (57%)	4/14 (29%)	5/14 (36%)	5/14 (36%)	2/14 (14%)
	2	LICA	1/2 (50%)	1/2 (50%)	1/2 (50%)	2/2 (100%)	2/2 (100%)
**75–79**	5	RICA	2/5 (40%)	2/5 (40%)	4/5 (80%)	1/5 (20%)	2/5 (40%)
	5	LICA	3/5 (60%)	3/5 (60%)	3/5 (60%)	4/5 (80%)	5/5 (100%)
**>80**	6	RICA	5/6 (83%)	2/6 (33%)	6/6 (100%)	3/6 (50%)	3/6 (50%)
	2	LICA	1/2 (50%)	1/2 (50%)	1/2 (50%)	2/2 (100%)	1/2 (50%)

**Table 6 diagnostics-12-02216-t006:** Pathways of collateral circulation in patients with bilateral ICA stenosis.

Age Group	Number of Patients	Stenosis	LECA	LVA	RECA	RVA
**65–69**	3	Bliateral	2	1	1	2
**70–74**	1	Bilateral	1	0	1	0
**75–79**	0	Bilateral	-	-	-	-
**>80**	2	Bilateral	0	0	0	2

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
