# Peer review of "Volumetric Flow Assessment in Extracranial Arteries in Patients with 70–99% Internal Carotid Artery Stenosis"

_diagnostics, 2022, doi:10.3390/diagnostics12092216_

Round 1

Reviewer 1 Report

Very interesting study and the results can help us follow up patients and also explain why some patients do better after cerebrovascular disorder than others. It can help us in the future for possible methods of treatment and prognosis using the collateral system and investigating how good the CBF is after stenosis or occlusion. Also volumetric flow assessment provides a useful tool to monitor and diagnose patients with cerebrovascular disorders. 

Author Response

Dear Sir or Madam,

On behalf of all authors I would like to thank you for the time and effort devoted to prepare the review, as well as for the positive opinion concerning our research. A minor changes have been implemented in the manuscript. Their list is presented below. 

The detailed list of changes in the manuscript.

  • lines 77-85 have been deleted
  • the “Table 1” number has been changed to the “Table 2”.
  • lines 170-172 have been deleted

Faithfully yours,

Piotr Kaszczewski

Reviewer 2 Report

-lines 77-85 should be deleted

-table 1 is presented twice

-“compensatory increased flow” or 114 “significant compensation.” Were defined as average + standard deviation from a reference study made but authors. However, if it was calculated in a different study, one should mind that the average population is in the range of confidence intervals. That means all values which fall between average +- 1.96 SD could be regular. Based on this, the authors should re-defined significant compensation as higher than average + 1.96 SD from the previous study. The same is for the definition of “decreased flow.” The rest of the manuscript should be corrected based on new outcomes

-please describe who the sonographer was and what is his experience

- the qualification to CAS or CEA is made on at least two USG examinations performed by two different physicians. It is due to the significant subjectivity of this study. Why did the authors not use the same protocol?

-line 165, please correct the p-value

-lines 170-172, please delete

-authors should present more clinical data regarding the patients included in the study

-what did the authors do when other vessels had stenoses like VA or ECA? Please present more detailed data

Author Response

Dear Sir or Madam,

On behalf of all authors I would like to thank you for the time and effort devoted to prepare the review as well as for the suggestions of implementing changes in the manuscript.

                I hope I will be able to clarify all issues addressed in the review report.

Referring to your questions concerning the experience of sonographers and study protocol, I would like to highlight that in our Department the patients who are qualified for surgery are examined by two different sonographers – according to the guidelines. Every patient who underwent surgery had 2 different Doppler ultrasound examinations performed preoperatively.

I would like to stress that this study was not focused on qualifying patients for the surgery (symptomatic/asymptomatic), but on assessment of the mechanisms and pathways of volumetric flow compensation. Our team had already published the study in which the correlation between diminished CBF values and the increased risk of occurrence of ischaemic symptoms were shown:

Kaszczewski, P.; Elwertowski, M.; Leszczyński, J.; Ostrowski, T.; Gałązka, Z. Volumetric Flow Assessment in Doppler Ultrasonography in Risk Stratification of Patients with Internal Carotid Stenosis and Occlusion. J. Clin. Med. 2022, 11, 531. https://doi.org/10.3390/jcm11030531

For the purpose of this study the patients who were diagnosed with 70-99% stenosis had also Doppler ultrasound examination with volumetric flow assessment in extracranial arteries performed. The volumetric assessment was performed by the same sonographer (Piotr Kaszczewski MD PhD – instructor of Doppler Ultrasonography in the Academy of European Society for Vascular Surgery  ESVS Academy, with the volume of over 40000 examinations performed). In the process of the qualification of patients to surgical treatment the examinations are performed also by MichaÅ‚ Elwertowski MD PhD (with the volume of almost 400 000 performed ultrasound examinations, including >30 000 Carotid ultrasound examinations).

Concerning setting the limit of compensation for average +/-1,96SD, there are several reasons of why we do not want to do it in the suggested way:

1.The clinical purpose of our whole research is to find a method that will allow to identify patients who will benefit haemodynamically from the surgery. Using our threshold we are able to identify such patients. The data were already presented in 2022 Charing Cross International Symposium in London and we hope to publish them after enlarging study group. Please, see the data on the graph presented below: (please see the word file attached).

<graph>

Using our thresholds of compensation we can clearly show that patients who fulfil our criteria of “significant compensation” do not have any changes of blood flow postoperatively. Patients with preoperative “no compensation” benefit most (postoperative flow increase of 366,2 ± 176,4 ml/min), while in patients with preoperative mild compensation the postoperative flow changes are prominent most (postoperative flow increase of 228,75 ± 136,9 ml/min) but less accentuated than in “no compensation group”.  

The abstract title was “A novel approach in the assessment of the influence of internal carotid stenosis treatment on cerebral blood flow volume changes in Doppler ultrasonography”. We are conscious that setting the compensation thresholds the way we did may cause questions, but we hope that the graph presented in this response letter will clarify your doubts and show that there is a purpose in such approach.

2. The current study, if accepted for publication, will be a forth study based on the criteria, which our team published. In order to make our current publication coherent and comparable to other our works we cannot make changes in “reference values”.

Referring to the issue of ECA and VA stenosis, in our study protocol which was published in our previous works, concomitant ECA or VA stenosis is listed among the exclusion criteria from the study group (Kaszczewski, P.; Elwertowski, M.; LeszczyÅ„ski, J.; Ostrowski, T.; GaÅ‚Ä…zka, Z. Volumetric Flow Assessment in Doppler Ultrasonography in Risk Stratification of Patients with Internal Carotid Stenosis and Occlusion. J. Clin. Med. 202211, 531. https://doi.org/10.3390/jcm11030531 and Kaszczewski, P.; Elwertowski, M.; Leszczynski, J.; Ostrowski, T.; Galazka, Z. Volumetric Carotid Flow Characteristics in Doppler Ultrasonography in Healthy Population Over 65 Years Old. J. Clin. Med. 20209, 1375. https://doi.org/10.3390/jcm9051375).

We aim to create a method of identifying patients with increased risk of ischaemic symptoms development, based on global cerebral inflow assessment, regardless of other risk factors (Global cerebral inflow volumetric assessment takes them into account). The fundamental aim of our study is confirmation and examination of collateral circulation and its pathways, which are independent of risk factors. 

Please find the detailed list of changes in the manuscript below.

  • lines 77-85 have been deleted
  • the “Table 1” number has been changed to the “Table 2”.
  • lines 170-172 have been deleted
  • Line 165 – p value of 0,005 is after Bonferroni The same value is in the table 4. It remained unchanged.

I hope I was able to clarify all the addressed issues, which will allow to consider our study suitable for publication in Diagnostics.

Faithfully yours,

Piotr Kaszczewski

Reviewer 3 Report

the paper contains too many limitations. Pharmacological, anthropometric, clinical characteristics might impact on results and reduce the strenght of the paper's conclusions. 

Author Response

Dear Sir or Madam,

On behalf of all authors I would like to thank you for preparing the review of our study.

I am afraid, you see our study in an unfavourable light, but I hope I will be able to clarify your doubts referring to our study protocol.

                The diseases that could possibly affect the CBF measurement were carefully taken into account, according to previously published study protocol. They include: stenosis of Common Carotid, External Carotid or Vertebral Artery, Concomitant diseases: uncontrolled hypertension, ischemic heart disease, heart insufficiency, positive history of heart infraction, positive history of stent implantation to coronary or any other arteries, cardiac arrhythmia, tachycardia, bradycardia, congenital vascular or heart failure, positive history of vascular interventions, presence of endocrine diseases: thyroid goiter, hyper-, hypothyroidism diabetes, adrenal diseases, positive history of thyroid surgery, smoking, alcohol use. Positive history of ischemic stroke, TIA symptoms or other neurological symptoms.

Our study protocol which has been published and used in our previous studies allows us to create homogenous study group in terms of factors which you listed.

I would also like to point out that volumetric cerebral inflow assessment may help to stratify the risk regardless of other concomitant disorders. For example even in patients after myocardial infraction the CBF may be preserved and maintained on unchanged level – our methods asses it.

The fundamental aim of our study is confirmation and examination of collateral circulation and it pathways, which are independent on other risk factors. 

As an example of the clinical purpose of our method, I would like to present you the data from our team study, titled “A novel approach in the assessment of the influence of internal carotid stenosis treatment on cerebral blood flow volume changes in Doppler ultrasonography”, which was presented this year on 2022 Charing Cross International Symposium in London. The clinical purpose of our whole research is to find a method that will allow to identify patients who will benefit haemodynamically from the surgery. Using our method we are able to do it. Please, see the data on the graph presented below (please see the word file):

<graph>

We can clearly show that patients who fulfil our criteria of “significant compensation” do not have any changes of blood flow postoperatively. Patients with preoperative “no compensation” benefit most (postoperative flow increase of 366,2 ± 176,4 ml/min), while in patients with preoperative mild compensation the postoperative flow changes are prominent most (postoperative flow increase of 228,75 ± 136,9 ml/min) but less accentuated than in “no compensation group”.  I hope we showed that there is a purpose in such approach.

Understanding mechanisms of collateral circulation may be of huge importance in diagnosing and monitoring patients with cerebrovascular diseases.

                A changes listed below has been made in the manuscript:

  • lines 77-85 have been deleted
  • the “Table 1” number has been changed to the “Table 2”.
  • lines 170-172 have been deleted

I hope my explanation will allow you to look at our study in a more favourable light.

Faithfully Yours,

Piotr Kaszczewski

Round 2

Reviewer 2 Report

The topic of establishing reference intervals is well described in the literature. I don't see any reason not to follow these guidelines and current scientific knowledge. The reference intervals are computed based on statistical means.  Authors are not allowed to adopt the lower and upper threshold limits based on their subjective opinion.  The properly calculated reference interval is crucial in the next stage - for subclassing patients as those with the increased or decreased flow.

The reference interval is estimated as a range of 95% values from healthy volunteers. It's mean +- 1.96SD. Authors should remember that the data should be normally distributed. Otherwise, the transformation should be performed. If data are not normally distributed, a nonparametric method can be used to calculate a 2.5th and 97.5th percentile range. 

Author Response

Dear Sir or Madam,

On behalf of all authors I would like to thank you for the second Review of our study. I hope I will be able to clarify your doubts concerning our reference values.

To start with, our team do not want to debate the well-established ways of creating norms. We are conscious that with normal distribution of data, the 95,45% of values are within 2 standard deviations from the mean and that with non-parametric distribution 2,5 and 97,5 percentiles should be calculated.

I would like to stress the fact that narrowing the confidence interval, when there is a clear cut-off point, is acceptable. What is more, it can in some situations provide the researcher with vital additional data and allow him to identify undiscovered correlations and connections. We do believe that after few years of preliminary research work, we have identified such cut-off point, which is located in the closest proximity of the one sigma distance from the mean. Therefore in 2020 our team published a study in which reference values were narrowed to one standard deviation from the mean.

To support my thesis I would like to highlight two facts:

  1. In the study “A novel approach in the assessment of the influence of internal carotid stenosis treatment on cerebral blood flow volume changes in Doppler ultrasonography” which was awarded in the 2022 Charing Cross International Symposium, using one sigma distance from the mean, we were able almost in 100%, who will benefit from carotid endarterectomy in terms of CBF increase, and who will not. What is more we were able to predict how large the CBF increase would be. This is applicable to all age groups older than 65 years old. The data are presented on the graph in our previous response letter.

Using two sigma distance almost all CBF values would be within 2-sigma reference values. Narrowing the confidence interval allowed us to show very strong correlation.

  1. In the study “Kaszczewski, P.; Elwertowski, M.; LeszczyÅ„ski, J.; Ostrowski, T.; GaÅ‚Ä…zka, Z. Volumetric Flow Assessment in Doppler Ultrasonography in Risk Stratification of Patients with Internal Carotid Stenosis and Occlusion.  Clin. Med.2022, 11, 531.” using the same one-sigma distance, we were able to show the correlation between the prevalence of ischaemic symptoms and diminished CBF. In symptomatic patients with ICA stenosis there is 2 times higher probability of encountering a person with diminished CBF, than in asymptomatic group. Again with 2-sigma, we will not be able to show the correlation, because almost all CBF values would be within 2 sigma reference values.

This proves that in our study narrowing the confidence interval to one standard deviation from the mean provided us with additional data about the patients.

It shows that dividing a patients into “no compensation”, “mild compensation” and “significant compensation” based on previously published reference values has sense, and there are statistically significant differences in the flow, flow patterns as well as the haemodynamic outcome of the surgery in all 3 groups.

The current study is a comparison of compensation mechanisms between the abovementioned groups.

Following amendments have been implemented in the manuscript body:

  • Referring to your remarks, the following paragraph was added in the manuscript body (lines: 431-437):

“We would also like to refer to the “reference values” which in our studies were initially defined within one standard deviation from the mean. During preliminary research our team identified a cut-off point in almost exactly one sigma from the mean. Not only does it allow us to identify differences between asymptomatic and symptomatic patients but also permits prediction of the postoperative increase in CBF in patients who undergo surgical treatment due to ICA stenosis [12,42]. Narrowing confidence interval is acceptable, when there are justified indications.”

  • Reference 42 was added: I hope I was able to clarify all the addressed issues, which will allow to consider our study suitable for publication in Diagnostics.

  • All comas were replaced by dots in decimals.

I hope I was able to clarify all the addressed issues, which will allow to consider our study suitable for publication in Diagnostics.

Faithfully yours,

Piotr Kaszczewski

Reviewer 3 Report

Please use dots rather than virgules for decimals.

Author Response

Dear Sir or Madam,

On behalf of all authors I would like to thank you for the second Review of our study.

Following amendments have been implemented in the manuscript body:

  • Referring to your remarks, the following paragraph was added in the manuscript body (lines: 431-437):

“We would also like to refer to the “reference values” which in our studies were initially defined within one standard deviation from the mean. During preliminary research our team identified a cut-off point in almost exactly one sigma from the mean. Not only does it allow us to identify differences between asymptomatic and symptomatic patients but also permits prediction of the postoperative increase in CBF in patients who undergo surgical treatment due to ICA stenosis [12,42]. Narrowing confidence interval is acceptable, when there are justified indications.”

  • Reference 42 was added: I hope I was able to clarify all the addressed issues, which will allow to consider our study suitable for publication in Diagnostics.

  • All comas were replaced by dots in decimals.

I hope I was able to clarify all the addressed issues, which will allow to consider our study suitable for publication in Diagnostics.

Faithfully Yours,

Piotr Kaszczewski
